# CoQ_10_ and Mitochondrial Dysfunction in Alzheimer’s Disease

**DOI:** 10.3390/antiox13020191

**Published:** 2024-02-02

**Authors:** Zdeněk Fišar, Jana Hroudová

**Affiliations:** Department of Psychiatry, First Faculty of Medicine, Charles University and General University Hospital in Prague, Ke Karlovu 11, 120 00 Prague, Czech Republic; hroudova.jana@gmail.com

**Keywords:** Alzheimer’s disease, coenzyme Q_10_, mitochondrial dysfunction, oxidative stress, drug

## Abstract

The progress in understanding the pathogenesis and treatment of Alzheimer’s disease (AD) is based on the recognition of the primary causes of the disease, which can be deduced from the knowledge of risk factors and biomarkers measurable in the early stages of the disease. Insights into the risk factors and the time course of biomarker abnormalities point to a role for the connection of amyloid beta (Aβ) pathology, tau pathology, mitochondrial dysfunction, and oxidative stress in the onset and development of AD. Coenzyme Q_10_ (CoQ_10_) is a lipid antioxidant and electron transporter in the mitochondrial electron transport system. The availability and activity of CoQ_10_ is crucial for proper mitochondrial function and cellular bioenergetics. Based on the mitochondrial hypothesis of AD and the hypothesis of oxidative stress, the regulation of the efficiency of the oxidative phosphorylation system by means of CoQ_10_ can be considered promising in restoring the mitochondrial function impaired in AD, or in preventing the onset of mitochondrial dysfunction and the development of amyloid and tau pathology in AD. This review summarizes the knowledge on the pathophysiology of AD, in which CoQ_10_ may play a significant role, with the aim of evaluating the perspective of the pharmacotherapy of AD with CoQ_10_ and its analogues.

## 1. Introduction

The neurodegeneration and aging of the brain are influenced by genetic and epigenetic factors, external and internal environment, lifestyle, trauma and diseases. Impaired synaptic and structural neuroplasticity is associated with pathophysiological, functional, and morphological changes in the brain that may serve as biomarkers of brain aging and neurodegeneration [1]. Due to the key role of mitochondria in bioenergetics, oxidative stress, metabolism, neuroinflammation, neuroplasticity, and apoptosis [2,3,4], attention has long been paid to mitochondrial dysfunction in aging and age-related neurodegenerative diseases such as Alzheimer’s disease (AD) [5,6,7]. Amyloid beta (Aβ)- and tau-induced neurotoxicity, which is linked to mitochondrial dysfunction and oxidative stress, have been postulated to play critical roles in the pathophysiology of AD [8].

In general, mitochondrial dysfunction is associated with reduced ATP production, increased production of reactive oxygen species (ROS), release of proapoptotic factors, and disturbed calcium homeostasis. The reduced production of ATP is mainly caused by the impaired function of the oxidative phosphorylation system (OXPHOS), which includes a series of redox reactions ending in oxygen, during which the mitochondrial electron transport system (ETS) transfers electrons between the complexes of the respiratory chain with the formation of the proton motive force, but also of superoxide [9]. The essential carrier of electrons from complex I or from complex II to complex III is coenzyme Q (CoQ), especially CoQ_10_ in humans. The availability and activity of CoQ_10_ and the Q-cycle play a significant role in the effectiveness of the OXPHOS system. In addition, CoQ_10_ functions as a lipid antioxidant and a necessary factor for controlling protein uncoupling and opening mitochondrial permeability transition pores (mPTPs).

CoQ_10_ is endogenously synthesized in every cell; it is also partially absorbed through food, which can affect its availability and activity. Supplementation of CoQ_10_ and its analogs is tested in the treatment of diseases related to oxidative stress and disruption of cellular energy, including AD and other neurodegenerative diseases [10].

This review summarizes the findings on the role of oxidative stress and mitochondrial dysfunction in the pathophysiology of AD, with a focus on the role of CoQ_10_. The aim of this review is to show that the understanding of the connection of specific Aβ and tau pathology in AD with mitochondrial dysfunction and oxidative stress, together with the consideration of the function of CoQ_10_, give the potential for further research into the role of CoQ_10_ and its analogues in the treatment of AD.

## 2. Alzheimer’s Disease

AD is a neurodegenerative disease defined as dementia confirmed by neuropathological observation of brain neuritic plaques formed by Aβ and neurofibrillary tangles (NFTs) composed of paired helical filaments of hyperphosphorylated tau protein (P-tau) [11,12]. The cause of the onset of AD, and how the neurotoxicity of Aβ and tau leads to the development of AD pathophysiology, are not yet sufficiently understood. In addition to amyloidopathy and tauopathy, mitochondrial dysfunction, oxidative stress, metabolic disturbances, neuroinflammation, disruption of neurotransmission, accumulation of transition metals, chronic hypoperfusion of brain tissue, and alterations in neurotrophic factors contribute to AD pathogenesis. The multifactorial nature of AD is captured in various interrelated biological hypotheses of AD (Figure 1) [13].

The main pathophysiological feature of AD is impaired proteostasis of pathways involved in the synthesis, folding, post-translational modifications, aggregation, targeting, and degradation of Aβ and tau protein in the brain. Mitochondria, mitochondria-associated membranes (MAMs), and endoplasmic reticulum (ER), which are connected to the ubiquitin proteasome system, autophagy system, ROS production, regulation of free cytosolic calcium, and apoptosis, are significantly involved in these processes [14,15,16,17].

The targets of new potential AD drugs are mainly processes related to Aβ and P-tau neurotoxicity, mitochondrial dysfunction, oxidative stress, metabolic disorders, and neuroinflammation [18,19,20].

### 2.1. Risk Factors

The exact cause of AD onset remains controversial. Progress is expected from longitudinal studies allowing for the identification of risk factors and early biomarkers of AD detectable in peripheral blood long before the onset of clinical symptoms of the disease [8,21]. The progress in understanding the pathophysiology of AD is linked to progress in the understanding of the mechanisms of aging-related neurodegeneration.

The main risk factor for the sporadic form of AD is age. Aging is a complex event, with gradual increase in cellular dysfunctions caused by the accumulation of protein, lipid, and nucleic acid dysfunctions. In general, age-related neurodegeneration is influenced by environmental factors (diet, exercise, and lifestyle), metabolic and oxidative stress, mitochondrial dysfunction, genetics and epigenetics, cerebrovascular dysfunction, blood-brain barrier dysfunction, neurotoxicity, and neuroinflammation [22,23].

Genetic variation significantly contributes to the risk of AD. Late-onset AD shows a heritability of 58–79% [24,25]. Occurrence of the ε4 allele of the gene *APOE* that encodes apolipoprotein E4 (ApoE4) is a major genetic risk factor for late-onset sporadic AD [26,27], as ApoE4 increases the neurotoxicity of Aβ and tau, which have a role in AD pathology [28]. Epigenetic changes, including changes in mitochondrial DNA (mtDNA), have been shown to be important in the pathogenesis of AD [29]. The autosomal dominant (familial) form of AD, which is defined as pathologically confirmed dominantly inherited AD, occurs in less than 1% of all cases [30]. However, all forms of AD are thought to share similar pathophysiological processes.

Other risk factors for AD are female gender [31,32], other genetic and epigenetic variations [26,33], brain injury [34], and internal and external environmental factors and stressors [35,36], including low levels of education [37], lifestyle [38], infection [39], cardiovascular disease [40], and metabolic dysregulation [41] such as type 2 diabetes mellitus (T2DM) [42]. The most significant environmental risk factors for the development of AD are late-life depression and T2DM [36].

### 2.2. Biomarkers

Validated biochemical biomarkers of AD are low concentrations of Aβ42 in cerebrospinal fluid (CSF), which reflect Aβ deposition in the brain, and increased tau in CSF, which is a marker of neuronal degeneration or damage. Tau biomarkers include both total tau (T-tau) and phosphorylated tau (P-tau) [43]. Neuroimaging biomarkers [44,45,46,47], proteomic and metabolomic markers [48], and biomarkers of oxidative stress [49], mitochondrial dysfunction [50,51], and neuroinflammation [52] are currently being sought and studied in AD.

The time course of measurable pathophysiological biomarkers in relation to the clinical course of AD can be used as a basis for the development of new drugs in AD that target pathophysiological processes in the early stages of the disease. A hypothetical time course of biomarker abnormalities and pathological changes in AD is proposed based on the longitudinal measurements of (i) Aβ and P-tau in CSF or in the brain by positron emission tomography (PET), (ii) neurodegeneration using fluorodeoxyglucose (FDG)-PET (hypometabolism) and magnetic resonance imaging (MRI) (hippocampal atrophy), (iii) synaptic dysfunction using FDG-PET and functional MRI (fMRI), (iv) neuroinflammatory changes, and (v) mitochondrial dysfunction [8,30,45,53,54]. According to this model, the onset of AD (even decades before the recognition of clinical symptoms of the disease) is associated with age-related mitochondrial dysfunction and the neurotoxicity of Aβ oligomers, while neurodegeneration and progression of AD is more associated with the neurotoxicity of P-tau oligomers and NFTs.

### 2.3. Mitochondrial Hypothesis

The mitochondrial cascade hypothesis of AD posits that mitochondrial dysfunction determines the initiation and development of this disease. According to this hypothesis, primary changes in mitochondrial function may induce a cascade of processes that lead to AD-specific neuropathological changes. Aβ and tau pathology/neurotoxicity may also be potentiated by interaction with mitochondrial proteins and membranes. According to the original mitochondrial cascade hypothesis [55], the basic level of mitochondrial function is genetically determined, and the decline of mitochondrial function is determined by aging processes, genetic factors, and environmental influences. If the decline in mitochondrial function exceeds a certain threshold, then the histological and pathophysiological changes specific to AD are triggered.

According to the mitochondrial cascade hypothesis, AD neuropathology arises secondary to mitochondrial dysfunction when the age-associated decline in mitochondrial function reaches a point where compensatory mechanisms are no longer effective [8,56,57]. The primary cause of the disease may not only be mitochondrial dysfunction, amyloidopathy or tauopathy, but also changes in the activity of factors that can cause them and which are localized in mitochondria, such as ApoE4 [58], glycogen synthase kinase 3 [59,60], and monoamine oxidase [61]. It appears that AD may have multiple initiating pathological factors that interact with each other.

Amyloid and tau pathology are specific for AD, but due to the mutual interactions and feedback effects of the aforementioned processes, it is not yet clear what triggers AD. According to the integrative amyloid-tau-mitochondrial hypothesis [8], the interaction of risk factors and biomarkers and their mutual synergy, rather than the primary effects of one particular factor, are decisive for the development of AD.

Mitochondrial dysfunction in AD includes bioenergetic impairment, increased ROS production, impaired mitochondrial dynamics and trafficking, and DNA mutations [62,63]. Mitochondrial abnormalities, including impaired function of mitochondrial ETS complexes and ATP production, have been described in AD [64]. One possibility to regulate the processes associated with neurodegeneration in AD is the regulation of the OXPHOS system through the availability and activity of CoQ_10_ using metabolic modulators, drugs, diet, and exercise [65,66].

The disruption of synaptic plasticity is one of the first steps in the neurodegeneration process associated with aging and the development of neurodegenerative diseases such as AD. Early deficits in synaptic mitochondria in AD include increased Aβ accumulation, mitochondrial dysfunction, increased mPTPs opening, decreased mitochondrial respiration, and decreased complex IV activity [67,68]. At the same time, Aβ and tau pathologies are in a reciprocal relationship with mitochondrial dysfunction and oxidative stress in AD [8,69,70,71]. Mitochondrial dysfunction associated with Aβ and tau pathology in AD includes decreased ATP production, mitophagy, biogenesis (decreased expression of peroxisome proliferator-activated receptor gamma coactivator 1-alpha, PGC-1α), activity of components of the OXPHOS system and other enzymes (e.g., 17β-hydroxysteroid dehydrogenase type 10, HSD-10), mitochondrial membrane potential (Δψ_m_), and import of mitochondrial proteins, imbalance of mitochondrial dynamics (upregulation of dynamin-like protein-1, DRP-1), impairment of intracellular Ca^2+^ homeostasis, membrane damage, interaction with adenine nucleotide translocator 1 (ANT1) and voltage-dependent anion channel 1 (VDAC1), and increased ROS production, apoptosis (increased cytochrome *c* release), and mPTP opening [8,63,68,72,73,74] (Figure 2).

In a mouse model of AD, the disruption of mitochondrial bioenergetics has been shown to precede the development of AD pathology [75]. Aβ accumulates in mitochondria and reduces the enzymatic activity of complex II and IV, reduces mitochondrial respiration, and impairs mitochondrial dynamics [73,76,77]. Damage to mitochondrial bioenergetics in AD was demonstrated both by measurements of the brains of AD transgenic mice and by PET neuroimaging in human AD brains [63]. Aβ and tau appear to act synergistically to damage the OXPHOS system, with tau damaging complex I and Aβ damaging complex IV [78].

### 2.4. Oxidative Stress Hypothesis

According to the oxidative stress hypothesis, the cause of the development of AD is oxidative stress, where damage to brain cells by ROS contributes to neurodegeneration and cognitive decline. Both mitochondrial dysfunction and Aβ and P-tau pathology contribute to increased ROS production [79]. ROS can trigger an inflammatory response and, conversely, inflammation induces oxidative stress [80]. Damage to synapses and brain cells due to oxidative stress may be both a consequence and a cause of Aβ and tau neurotoxicity in AD [81].

The oxidative stress hypothesis is based on the observation that oxidative stress, i.e., an imbalance between the production and elimination of ROS, occurs in neurodegeneration associated with AD. Increased ROS production and/or reduced antioxidant defenses lead to the accumulation of dysfunctional proteins, lipids and nucleic acids (including mtDNA) and impaired mitophagy.

In neurons, superoxide is mainly produced in the mitochondrial matrix during electron transfer in ETS (mainly generated by the mitochondrial complexes I and III). Superoxide is converted directly in the matrix by superoxide dismutase to less reactive hydrogen peroxide, which, however, passes through membranes and can be converted to a very reactive hydroxyl radical in the cytosol [9]. This radical can initiate increased peroxidation of membrane lipids in AD [82]. Lipophilic antioxidants such as CoQ_10_ have a protective effect on the peroxidation of membrane lipids; thus, administration of CoQ_10_ and/or regulation of CoQ_10_ activity in the ETS is being tested in AD therapy.

Mitochondrial dysfunction can lead to both oxidative and nitrosative stress, as impaired electron transfer in the mitochondrial ETS leads to increased production of ROS, and impaired transport of calcium ions into the mitochondrial matrix induces increased production of nitric oxide (NO) and the formation of reactive peroxynitrite.

## 3. Coenzyme Q_10_

CoQ_10_ (ubiquinone, 2,3 dimethoxy-5-methyl-6-decaprenyl-1,4-benzoquinone) is the most common form of CoQ in humans [83]; it is found in all cell membranes, where it acts as an electron carrier and antioxidant. The internal synthesis of CoQ_10_ starts from the amino acid tyrosine (benzoquinone ring) and from the mevalonate pathway (isoprenoid side chain) (Figure 3) [84,85]. Current knowledge about CoQ_10_ biosynthesis is described in detail in a series of reviews [86,87,88]. A collection of enzymes that produce CoQ_10_ (termed as complex Q) are localized in the inner mitochondrial membrane (IMM) and in the ER [89,90].

The main cellular functions of CoQ_10_ include (i) electron transport in the mitochondrial ETS, with a key role in the formation of ATP, (ii) antioxidant action, including protection against lipid peroxidation, participation in the reduction/recycling of other antioxidant molecules (α-tocopherol and ascorbate), and stabilization of the plasma membrane and cell redox balance, (iii) apoptosis and modulation of mPTPs, (iv) signaling modulation of gene expression, including anti-inflammatory effects, (v) maintenance of the proton gradient on the lysosomal membrane, and (vi) activation of mitochondrial uncoupling proteins [90,91,92,93,94,95,96,97].

CoQ_10_ is present in mitochondria as part of the electron transport chain. CoQ_10_ is a lipophilic molecule located in cell membranes, mainly in the IMM. It is crucial for electron transfer between complex I or II and complex III of the respiratory chain, and also for electron capture from dehydrogenases [10,98,99] localized on the outer or inner surface of the IMM. Moreover, CoQ_10_ and its redox state could indirectly modulate a number of mitochondrial and non-mitochondrial metabolic pathways, such as sulfide metabolism, one-carbon metabolism, glutathione, and ferroptosis; therefore, it can be included in the pathophysiology of some metabolic diseases [99].

CoQ_10_ occurs in three redox forms: as an oxidized form (ubiquinone, C_59_H_90_O_4_), a reduced form (ubiquinol, C_59_H_92_O_4_), and a semi-oxidized form (semiquinone, C_59_H_91_O_4_^●^; under physiological conditions it occurs as an ion C_59_H_90_O_4_^–●^) (Figure 3). The reduced form of CoQ_10_(H_2_) (ubiquinol) acts as an antioxidant and scavenger of free radicals, thereby preventing oxidative damage to lipids, proteins and nucleic acids. On the other hand, CoQ_10_ can also have a pro-oxidant role, as it can participate in the formation of superoxide and hydrogen peroxide [100].

### 3.1. CoQ_10_ in OXPHOS System

High-energy electrons enter the mitochondrial ETS via complex I (from the reduced coenzyme nicotinamide adenine dinucleotide, NADH) or via complex II (from the reduced coenzyme flavin adenine dinucleotide, FADH_2_). The product of redox reactions in the complex I is the transfer of 4 protons per 1 NADH molecule into the intermembrane space. Electrons are transferred from complex I or from complex II to complex III (coenzyme Q:cytochrome *c*—oxidoreductase, sometimes called the cytochrome *bc_1_* complex) by ubiquinol (reduced CoQ_10_(H_2_)) and enter the protonmotive Q-cycle of the complex III. In the Q-cycle mechanism, protons are translocated across the IMM as a result of the reoxidation of ubiquinol at the Q_o_-site on the outer side of the IMM, and the reduction of ubiquinone (CoQ_10_) at the Q_i_-site on the matrix side of the IMM (Figure 4). The respiration subunits in the Q-cycle are cytochrome *c*_1_ (cyt *c*_1_), cyt *b* (with low (*b_L_*) and high (*b_H_*) potential hemes), and Rieske protein. The substrates of redox reactions catalyzed by complex III are ubiquinol and two molecules of ferricytochrome *c*, and the products are ubiquinone, two molecules of ferrocytochrome *c*, and four protons (released into the intermembrane space and used by ATP synthase during ATP biosynthesis) [92,101,102,103]. Total mitochondrial ETS activity is most easily measured as the kinetics of mitochondrial oxygen consumption [104,105]. Note that the Q_o_-site of the complex III is an important site for generation of superoxide, and thus has a role in oxidative damage during aging and neurodegeneration.

The efficiency of the OXPHOS system, especially the transfer of electrons by means of CoQ_10_ and the Q-cycle, is influenced by the existence and organization of respiratory supercomplexes [106,107]. The most common supercomplexes (respirasomes) observed are complex I/III, complex I/III/IV, and complex III/IV. Supercomplex assembly is dynamic, and their formation and stabilization depend on the lipid composition of the IMM, especially on the presence of cardiolipin [108]; the initiation of their formation can be associated with the membrane potential. The formation of supercomplexes can significantly increase the efficiency of the OXPHOS system by optimizing the utilization of ETS substrates, stabilizing complex I and reducing the formation of ROS [109,110]. A constitutive part of respirasomes is CoQ_10_, which exists in different CoQ_10_ pools in the IMM [111].

The dependence between respiratory complexes, supercomplex assembly dynamics, and the existence of CoQ_10_ pools in the effectiveness of the ETS system is intensively studied [112]. According to the classic fluid model, respiratory complexes I–IV are randomly distributed in the IMM, and electron transfer between them is realized by electron carriers and CoQ_10_ located inside the IMM and cyt *c* located on the external surface of the IMM (Figure 4). The random collision model [113] assumes that mitochondrial electron transport is a process of random collisions based on the diffusion of individual components in the fluid IMM. The discovery of the existence of supercomplexes in the respiratory chain [114] and the recognition of their function [115] led to the modification of the fluid model to the plasticity model, which adds to the fluid model a new view of the structural and functional complexity in the transfer of electrons in the mitochondrial ETS, as well as the function of the CoQ_10_ pool in the IMM [111,116,117].

The plasticity model assumes that respiratory complexes can function either individually or as components of supercomplexes. It has long been assumed that free CoQ_10_ resides in the IMM in a homogeneous pool [118]. With the discovery of respirasomes and their function, it is shown that there is a segmentation of CoQ_10_ molecules, into a pool attached by supercomplex I/III (CoQ_NADH_ pool) and into a pool available for complex II and other enzymes (CoQ_FADH_ pool) [109], while the two pools interact with each other. The majority of CoQ that receives electrons from FADH_2_ and the majority of cyt *c* apparently remain unbound to supercomplexes.

### 3.2. CoQ_10_ in AD Pathophysiology

CoQ_10_ acts primarily as an antioxidant and bioenergetic modulator (Figure 2). Its antioxidant, neuroprotective, and anti-inflammatory effects can improve oxidative stress and neuroinflammation associated with AD [119]; its effects on the efficiency of the mitochondrial ETS system [120] can contribute to the elimination or prevention of mitochondrial dysfunction in AD.

It is accepted that mitochondrial dysfunctions in the pathophysiology of AD involve the altered activities of several mitochondrial enzymes, including those in the OXPHOS system, mainly in the reduced activity of complex IV [121,122,123]. The role of CoQ_10_ in bioenergetics led to the assumption that the availability and activity of CoQ_10_ may contribute to impaired energy metabolism in AD.

One of the mitochondrial parameters that characterizes the efficiency of the OXPHOS system (including the efficiency of CoQ_10_) is the kinetics of oxygen consumption. Mitochondrial respiratory capacity in platelets from AD patients was confirmed to be significantly reduced and negatively correlated with cognitive decline, as measured using the Mini Mental State Examination (MMSE) score [50,124]. At the same time, an association between MMSE score and plasma CoQ_10_ concentration in AD patients was revealed [51].

CoQ_10_ may play a role in specific AD pathological processes. This is supported by protective effects of CoQ_10_ against Aβ neurotoxicity [125] and the observation that serum CoQ_10_ levels may be inversely associated with dementia risk [126]. A role for CoQ_10_ in AD is supported by a study in which the in vitro induction of tau aggregation by CoQ and its presence in paired helical filaments was observed [127].

It is supposed that the determination of serum CoQ_10_ levels could be useful for predicting the development of AD. A systematic review and meta-analysis of studies measuring CoQ_10_ levels in AD patients and other dementias, and a review of the therapeutic effects of CoQ_10_ in humans with AD, was recently published [128] and concluded that that CoQ_10_ deficiency has not been sufficiently demonstrated in peripheral blood of AD patients.

Oxidative stress is also thought to contribute to AD progression by inducing Aβ overexpression and accumulation, while it is unclear what is the primary cause of disease development. Oxidative stress modulates proteostasis, which is strongly impaired in AD [129]. Therefore, antioxidants are among the new clinically tested potential drugs in AD [20]. The reduced form of CoQ_10_(H_2_) is a known antioxidant that also affects Aβ pathology [130]. The protective effects of CoQ_10_ and other antioxidants against Aβ accumulation in the brain, and against neuroinflammation and hypoxia, have been reported [131].

The therapeutic importance of CoQ_10_ in the treatment of AD has mainly been demonstrated by studies in cellular and animal models of AD, but has not been confirmed in humans. The neuroprotective effect of CoQ_10_ on cognitive impairments and oxidative damage of specific areas of the brain of a rat was established [132]. A study with a mouse model of AD showed that there is a relationship between changes in the hippocampus and cerebral cortex and oxidative stress, proteostasis, and bioenergetics, and that CoQ_10_ may act preventively [133]. Cell culture studies have shown a protective effect of CoQ_10_ on Aβ neurotoxicity [125,130,134].

The administration of CoQ_10_ as an antioxidant had no significant effect on biomarkers associated with amyloid and tau pathology in AD measured in CSF (Aβ_42_, tau, and P-tau), nor on cognitive function [135]. Also, the administration of CoQ_10_ did not have a significant therapeutic effect in other neurodegenerative diseases such as Parkinson’s disease and Huntington’s disease [136]. Currently, the changes in CoQ_10_ levels in AD patients, and their effects on improving neuropsychological assessments, are not clearly confirmed, and the exact role of CoQ_10_ in the pathogenesis of AD and the treatment of neurodegenerative diseases is yet to be clarified [91,128]. Studies that provide clinical and mechanistic data regarding the efficacy of CoQ_10_ in the treatment of AD-related cognitive decline are lacking. The results so far indicate a lower risk of cognitive impairment in those with normal cognition who consumed CoQ_10_ daily; therefore, supplementation of CoQ_10_ can be recommended to slow cognitive decline in AD [137].

## 4. AD Treatment

### 4.1. Targets of Novel AD Drugs

The drugs currently approved or recommended for the treatment of AD belong to the category of agents targeting neurotransmitter receptors (cholinesterase inhibitors and *N*-methyl-d-aspartate receptor antagonists) and Aβ (monoclonal antibodies directed against Aβ plaques, protofibrils, and oligomers). In clinical use, there are drugs aimed at suppressing or alleviating the symptoms of AD (donepezil, rivastigmine, galantamine, memantine and the memantine/donepezil combination). Two disease-modifying drugs (DMDs) targeting Aβ pathology (aducanumab and lecanemab) have recently received accelerated approval from the United States Food and Drug Administration (FDA). Attention is mainly devoted to the development of new effective and specific DMDs. However, there is also ongoing testing of symptomatic substances aimed at improving cognitive functions and neuropsychiatric symptoms in AD, and these are often drugs already approved for the treatment of other diseases or substances used in alternative and complementary medicine [138].

The categories of biological processes and AD drug targets that are closely related to mitochondrial function (metabolism and bioenergetics, synaptic plasticity/neuroprotection, and cell death) and oxidative stress are included in the CADRO classification system (“Common Alzheimer’s and Related Dementias Research Ontology”). According to the periodic annual review of AD drug development [20], a total of 141 agents were tested for the treatment of AD and mild cognitive impairment (MCI) in phase 1, 2, or 3 clinical trials at the beginning of 2023. They mainly targeted inflammation (17.0%), Aβ (15.6%), synaptic plasticity/neuroprotection (12.9%), tau (9.2%), metabolism and bioenergetics (7.1%), and oxidative stress (5.0%).

In addition to new DMDs, drugs targeting the primary causes of AD onset and appropriate combinations of approved drugs with adjuvant agents are also being sought and tested. These supplements, such as CoQ_10_, ω-3 fatty acids, soy, ginkgo biloba, B vitamins, vitamin D plus calcium, vitamin C, and β-carotene have not yet been shown to prevent cognitive dysfunction in AD [128,139]. It can be expected that if adjuvants have anti-amyloid, anti-tau, neurochemical, mitochondrial, antioxidant or anti-inflammatory effects [125,140], they may also have the therapeutic potential to moderate the progression of cognitive impairment in AD.

Drugs in phase 2 or 3 clinical trials that target oxidative stress include hydralazine, icosapent ethyl, polyunsaturated fatty acids (PUFA), omega-3, edavarone, and Flos gossypii flavonoids. Note that, although oxidative stress is accepted as a key modulator of the biological processes in neurodegenerative diseases [141], therapeutic interventions must also take into account the useful endogenous mechanisms regulated by ROS [142]. Metabolism and bioenergetics are targeted by metformin, tricaprilin, insulin intranasal (+empagliflozin), dapagliflozin, empagliflozin, semaglutide, T3D-959, Chinese traditional medicine, and obicetrapib [20].

Currently, antioxidants are used as supplements to approved AD drugs. The lack of effectiveness of antioxidants in AD therapy can be explained by their non-specific intervention in the balance between ROS activity and the activity of the antioxidant system, which can suppress the useful and necessary role of free radicals in certain areas of the brain. Nevertheless, the testing of CoQ_10_ and its analogues as a supportive therapy for AD remains important, primarily in the context of suppressing excessive peroxidation of membrane lipids and enhancing bioenergetics.

A suitable target for substances that restore or increase mitochondrial function is the stimulation of mitochondrial biogenesis. Mitochondrial biogenesis is associated not only with cell division, but also with the response to oxidative stress (that is, with the demand for increased cellular energy consumption), exercise, hormones, electrical stimulation, etc. Mitochondrial biogenesis has become the target of new drugs for diseases associated with mitochondrial dysfunction, including neurodegenerative disorders such as AD. Pathways associated with mitochondrial biogenesis, and activated in response to energy deficit, include the activation of the peroxisome PGC-1α axis, activation of the AMP-activated protein kinase (AMPK), and Sirtuin 1 activation [63].

In summary, the main approach in the search for new DMDs for AD is the targeting of the pathophysiological processes that cause the onset of the disease, which, according to current AD hypotheses, are Aβ and tau pathology, mitochondrial dysfunction, oxidative stress, neuroinflammation, disturbed neurotransmission, and disorders of brain metabolism. If we assume that age is the main risk factor for AD, then we can focus on the regulation of processes and biomarkers discussed in the oxidative stress hypothesis and in the mitochondrial hypothesis.

### 4.2. CoQ_10_ in AD Treatment

Substances that act as antioxidants or increase mitochondrial bioenergetics have potential for the treatment of neurodegenerative diseases such as AD. From the perspective of the mitochondrial hypothesis of AD and the possibility of pharmacologically influencing mitochondrial dysfunction, the synaptic plasticity and metabolism of brain cells, CoQ_10_, and mitochondrial proteins and lipids interacting with it also appear to be potential targets for new AD drugs. These substances are capable of regulating the availability and activity of CoQ_10_, and thus the function of the OXPHOS system.

CoQ_10_ is mainly formed endogenously, so the cause of a deficiency in CoQ_10_ can mainly be its impaired biosynthesis, increased degradation, or increased usage. In mitochondrial diseases, CoQ_10_ deficiency can primarily be caused by mutations in the genes responsible for CoQ_10_ biosynthesis, or secondarily by defects in other genes [143]. Endogenous metabolites, which are also involved in cholesterol production, may be involved in the regulation of CoQ_10_ biosynthesis [90]. By upregulating the synthesis of CoQ_10_, not only its concentration but also its appropriate mitochondrial localization can be achieved. However, dietary intake can also contribute to CoQ_10_ availability, especially when its endogenous production is reduced with aging or with genetic mutations primarily or secondarily involved in CoQ_10_ biosynthesis.

Supplemental CoQ_10_ is well tolerated, but because it is almost insoluble in water, its therapeutic use is limited. To minimize absorption and increase plasma levels of CoQ_10_, it is recommended to administer it in multiple doses and while dissolved in an oil matrix [144]. To improve the bioavailability of CoQ_10_, solubilized formulations, including water-soluble CoQ_10_ analogues, have been developed [145,146,147,148,149,150]. On the periphery, CoQ_10_ is transported in lipoproteins [151]. The bioavailability and efficacy of oxidized and reduced forms of CoQ_10_ are similar [152]. CoQ_10_ has been confirmed to cross the blood-brain barrier in rats and mice, and its oral or intravenous administration leads to increased levels of CoQ_10_ in the brain [151,153,154], brain mitochondria [155], and an increase in the number of mitochondria in the brain [156], which result in an increase in cellular antioxidant potential [157] and bioenergetics [158].

CoQ_10_ and other antioxidants or bioenergetics stimulators may be potentially effective in the treatment of neurodegenerative diseases [159,160]. CoQ_10_ is well characterized as a neuroprotective antioxidant in animal models and in human studies of neurodegenerative disorders. CoQ_10_ has been shown to reduce oxidative stress and amyloid pathology in a mouse model of AD [161]. Age-related decline in mitochondrial function has been shown in a mouse model to be accompanied by decreased levels of mitochondrial CoQ, and exogenous administration of water-soluble CoQ can lead to the restoration of mitochondrial function [148]. A number of studies have confirmed the significant neuroprotective effects of CoQ_10_ in experimental biological models, but the suitability of CoQ_10_ as a biomarker or drug has not been confirmed in AD patients [128].

Given the role of CoQ_10_ in bioenergetics and antioxidant activity, and the observation that CoQ_10_ has protective effects against Aβ-induced cell toxicity and impaired synaptic plasticity [134,162], CoQ_10_, and the processes regulated by it, have strong therapeutic potential in AD [163]. Although studies in animal models of AD show significant improvements in cognition, clinical trials have not been very successful. Therefore, upregulation of brain CoQ_10_ biosynthesis appears to be more suitable for the treatment of neurodegenerative disorders. An increase in CoQ_10_ biosynthesis is possible in a physiological way (cold adaptation and exercise) [84], or by targeting of isoprenoid regulation within mevalonate pathway [164], e.g., by the administration of substances, such as epoxidized all-trans polyisoprenoids [90].

The reduced availability of CoQ_10_ can be eliminated by the dietary administration of this substance. The process of CoQ_10_ absorption and bioavailability is complex and strongly depends on the formulation of the preparation [152]. CoQ_10_ and its analogues, idebenone and mitoquinone (MitoQ), are used in the treatment of mitochondrial disorders and in the supportive treatment of neurodegenerative diseases associated with mitochondrial dysfunction, such as AD [85]. CoQ_10_ is well tolerated and safe, but not approved for the treatment of AD [165].

The synthetic analogue of CoQ_10_ idebenone (hydroxydyecylubiquinone) acts as an antioxidant and an electron carrier in ETS. Idebenone has protective effects against many toxins [166], but inhibits complex I [167]. In some studies it showed therapeutic effects on AD progression [168], but in other studies no effect of idebenone on cognitive decline in AD [169] or on biomarkers related to Aβ and tau pathology in AD was demonstrated [135].

The administration of MitoQ, which passes through membranes more easily and concentrates in the mitochondria thanks to the attached triphenylphosphonium to ubiquinone, seems promising. MitoQ has protective effects against mitochondrial damage [170] and appears promising in suppressing AD symptoms. In a Caenorhabditis elegans model, the administration of MitoQ extends lifespan and shows protective effects against Aβ toxicity [171]. In experiments with cell cultures and a mouse model of AD, MitoQ was found to increase synaptic connectivity and neurite outgrowth, prevented Aβ-induced oxidative stress and improved memory retention [172,173,174].

In summary, the mitochondria-targeted CoQ_10_ analogs appear promising for AD therapy. To evaluate the effect of CoQ_10_ on the activity of the OXPHOS system, or on mitochondrial dysfunction associated with impaired electron transport in the ETS in AD, data from in vivo measurements are not yet available. However, in vitro measurements using isolated mitochondria suggest that exogenously supplied CoQ_10_ can increase the mitochondrial respiratory rate. It can be hypothesized that the antioxidant activity of CoQ_10_, which protects mitochondrial membranes from oxidative damage, and the increase in electron transfer efficiency due to the incorporation of CoQ_10_ into the IMM, contribute to this increase in ETS efficiency. From this point of view, it appears as a perspective synthesis and testing of (i) analogues of CoQ_10_ with good bioavailability in brain mitochondria, (ii) substances aimed at regulating CoQ_10_ biosynthesis, and (iii) substances aimed at increasing the activity of CoQ_10_ in the Q cycle, including those that affect the formation of supercomplexes containing CoQ_10_ and complex III.

## 5. Discussion and Conclusions

Based on the findings on which the mitochondrial hypothesis of AD and the oxidative stress hypothesis are based, mitochondrial ETS, especially CoQ_10_ and the processes mediated by it, appear to be a promising target for new AD drugs. The uptake of dietary CoQ_10_ into tissues is limited, as CoQ_10_ is localized in the membranes of the central hydrophobic part of the lipid bilayer; thus, the space available for CoQ_10_ and similar lipophilic compounds is limited. CoQ_10_ diffusion in the lipid bilayer may represent the rate-limiting step of electron transfer [175].

CoQ_10_ concentration decreases with aging, with the availability and redox status of CoQ_10_ playing a role in the oxidative stress associated with aging [176,177,178,179]. The increased availability of CoQ_10_ may have neuroprotective effects through antioxidant and bioenergetic effects. However, a systematic review did not confirm a decrease in plasma CoQ_10_ in AD patients [128]. Also, the decrease in mitochondrial respiratory rate in platelets with age was not different in AD patients compared to age-matched healthy controls [180]. These results indicate that mitochondrial dysfunction (potentially associated with reduced availability and activity of CoQ_10_) in AD is associated with the onset of the disease rather than its progression. CoQ_10_ supplementation may then slow down disease-related neurodegeneration.

When we tested in vitro the effect of CoQ_10_ and other antioxidants on mitochondrial respiration using a model of isolated brain mitochondria, only the addition of CoQ_10_ caused an increase in the respiratory rate [120]. These results demonstrate that it is possible to regulate OXPHOS activity by CoQ_10_. It can be hypothesized that the increase in ETS activity may be achieved by the direct incorporation of CoQ_10_ into the IMM, rather than the antioxidant action of CoQ_10_. It can be assumed that the improvement of mitochondrial function in AD is possible by increased CoQ_10_ biosynthesis and increased mitochondrial biogenesis, rather than by dietary CoQ_10_ supplementation, which does not ensure an increase in CoQ_10_ in IMM of brain mitochondria. To increase the availability of CoQ_10_ in the brain and in brain mitochondria, its exogenous administration and stimulation of biosynthesis can be combined. Exogenous supplementation of CoQ_10_ is safe and does not affect its endogenous biosynthesis [181].

Age-associated mitochondrial dysfunction (measurable as a decrease in ETS capacity or a decrease in respiratory reserve), together with the effects of ApoE4, may be at the start of Aβ and tau pathology, oxidative stress, metabolic dysregulation, and neuroinflammation in the late-onset sporadic form of AD (Figure 2). Targeting new AD drugs on the activity and efficiency of ETS, specifically on the availability and activity of CoQ_10_ in the IMM and on the regulation of redox processes in the ETS associated with CoQ_10_, can therefore be considered a promising research approach. The direct in vitro effects of CoQ_10_ on increasing mitochondrial respiration [120] suggest that the regulation of CoQ_10_ biosynthesis could be a promising direction in the development of new AD drugs.

In conclusion, the specific pathophysiology of AD is primarily associated with Aβ and tau pathology. According to the mitochondrial hypothesis and the oxidative stress hypothesis, mitochondrial dysfunction and oxidative stress are involved in the pathogenesis of AD, which can be both the initiating and accompanying processes in the pathophysiology of AD and the development of neurodegenerative processes in AD. The approach of targeting new AD drugs to the availability and activity of CoQ_10_ is underpinned by the role of CoQ_10_ in the cellular antioxidant system and in mitochondrial bioenergetics. Increasing the availability and activity of CoQ_10_ is possible by its exogenous administration. Using biological models of isolated brain mitochondria, cell cultures, and animal models of AD, both the antioxidant and bioenergetic effects of CoQ_10_ have been demonstrated. However, the effects of antioxidants are shown to be insufficiently effective in AD therapy in humans, and exogenous administration of CoQ_10_ does not yet allow its reliable increased utilization by brain mitochondria. From this point of view, targeting new AD drugs to increase mitochondrial bioenergetics by regulating mitochondrial biogenesis or CoQ_10_ biosynthesis in the brain appears to be a more appropriate pharmacological strategy. The regulation of the activity of the OXPHOS system through increasing the efficiency of electron transfer in the ETS using CoQ_10_ and cyt *c* appears promising. However, targeted pharmacological intervention in mitochondrial electron transfer requires a deeper understanding of the normal and pathological processes in the OXPHOS system, including those associated with the assembly of respiratory complexes into respirasomes and the function of supercomplexes and CoQ_10_ in electron transport efficiency.

Considering the role of CoQ_10_ in bioenergetics and lipid peroxidation, it is advisable to continue studying the possibilities of AD therapy by regulating the activity of CoQ_10_ in the OXPHOS system. In the early stages of AD development, the stimulation of mitochondrial bioenergetics and the antioxidant action of CoQ_10_ could prevent the development of Aβ and tau neurotoxicity. However, even in the later stages of the disease, the effects of CoQ_10_ on mitochondrial bioenergetics could slow the progression of the disease. Insights into the possible role of CoQ_10_ in the pathophysiology of AD show that the regulation of mitochondrial function using CoQ_10_ and its analogues is a promising approach in the development of new AD drugs and protection against Aβ- and tau-induced neurotoxicity.

## Figures and Tables

**Figure 1 antioxidants-13-00191-f001:**
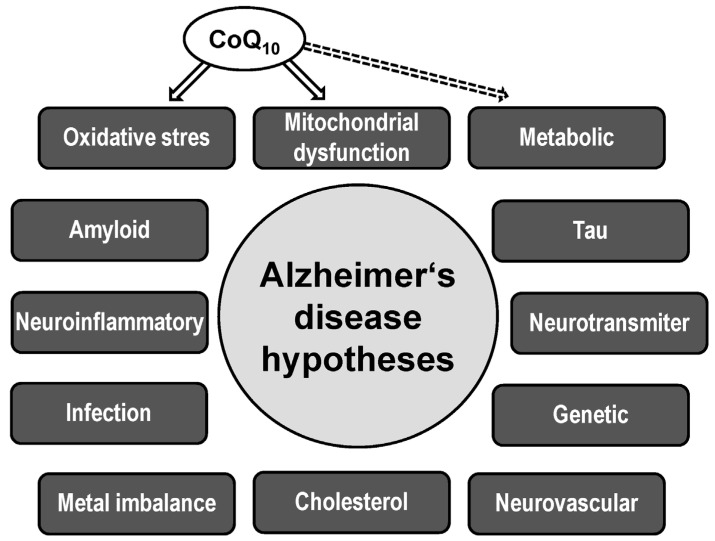
Biological hypotheses of Alzheimer’s disease. Solid arrow shows direct effect, dotted arrow shows indirect effect of coenzyme Q_10_ (CoQ_10_).

**Figure 2 antioxidants-13-00191-f002:**
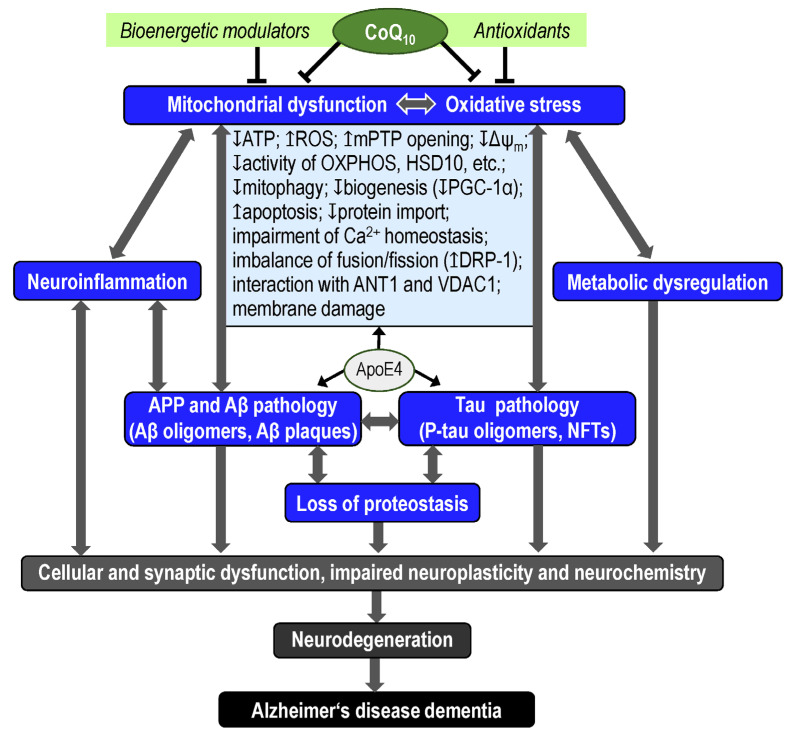
Simplified scheme of Alzheimer’s disease (AD) pathophysiology, with focus on the role of mitochondrial dysfunction. AD pathophysiology is associated with amyloid beta (Aβ) pathology (neurotoxicity of Aβ oligomers and plaques), tau pathology (neurotoxicity of tau oligomers and neurofibrillary tangles), mitochondrial dysfunction, oxidative stress, neuroinflammation, and loss of proteostasis. All these processes are interrelated and result in cellular and synaptic dysfunction, impaired neuroplasticity and neurochemistry, neurodegeneration (synaptic and neuronal loss and brain atrophy), cognitive decline, and AD dementia. Mitochondrial dysfunction associated with Aβ and tau pathology in AD includes decreased ATP production, mitophagy, biogenesis (peroxisome PGC-1α), activity of components of the OXPHOS system and other enzymes, mitochondrial membrane potential (Δψ_m_), and import of mitochondrial proteins, imbalance of mitochondrial dynamics (DRP-1), impaired of intracellular Ca^2+^ homeostasis, membrane damage, interaction with ANT1 and VDAC1, and increased ROS production, apoptosis, and mPTP opening. Coenzyme Q_10_ (CoQ_10_), as a unique endogenous antioxidant and electron transporter in the OXPHOS system, may have a significant role in the pathophysiology and treatment of AD, primarily through the regulation of mitochondrial function. ANT1—adenine nucleotide translocator 1; ApoE4—apolipoprotein E4; APP—amyloid precursor protein; DRP-1—dynamin-like protein-1; HSD10—17β-hydroxysteroid dehydrogenase type 10; mPTP—mitochondrial permeability transition pore; NFTs—neurofibrillary tangles; OXPHOS—oxidative phosphorylation; PGC-1α—peroxisome proliferator-activated receptor gamma coactivator 1-alpha; P-tau—phosphorylated tau; ROS—reactive oxygen species; VDAC1—voltage-dependent anion channel 1.

**Figure 3 antioxidants-13-00191-f003:**
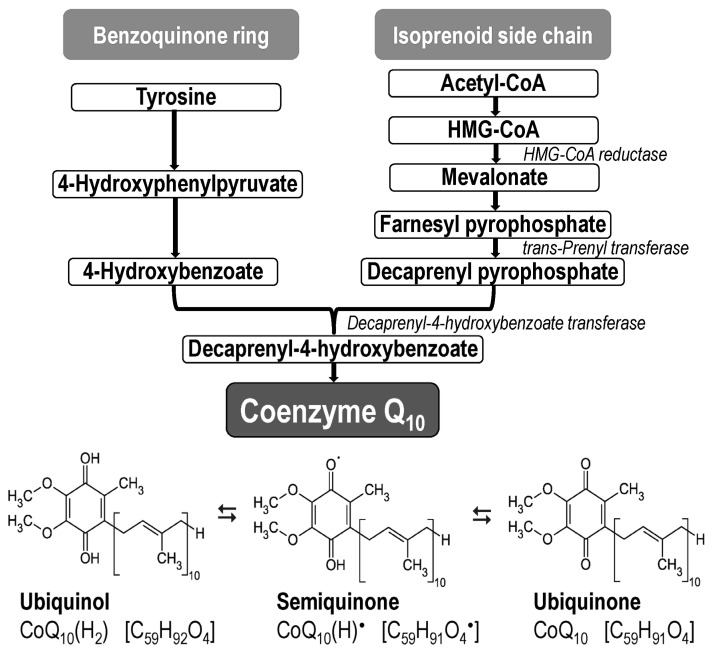
Basic steps in coenzyme Q_10_ biosynthesis and three redox isoforms. HMG-CoA—β-Hydroxy β-methylglutaryl-coenzyme A.

**Figure 4 antioxidants-13-00191-f004:**
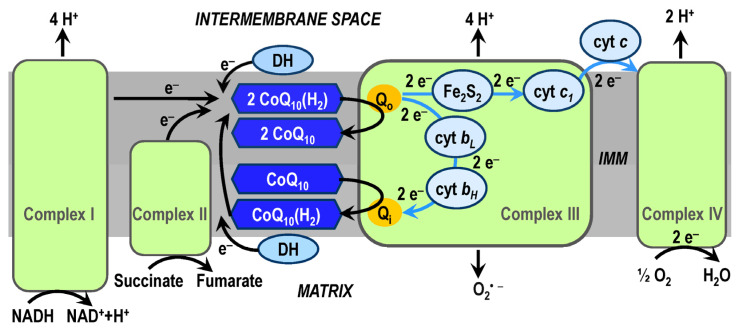
A simplified diagram of the mitochondrial electron transport system (ETS) with electron transfer in the Q-cycle of complex III. Electrons enter the ETS via complex I or via complex II and are transferred by coenzyme Q_10_ (CoQ_10_) to complex III. CoQ_10_ can also transfer electrons from dehydrogenases (DH) located on the outer or inner surface of the inner mitochondrial membrane (IMM). In the Q_o_-site complex III near the outer side of the IMM, two electrons from one ubiquinol (CoQ_10_(H_2_)), and subsequently two more electrons from the second ubiquinol, pass into two bifurcated transfer chains: (1) The acceptor of the two electrons is the iron –sulfur cluster (Fe_2_S_2_ center) of the Rieske protein, which passes electrons via cytochrome *c*_1_ (cyt *c*_1_) to cytochrome *c* (cyt *c*) on the outer surface of the IMM. Cyt *c* transfers electrons to complex IV (cytochrome *c* oxidase), where oxygen is finally reduced to water. (2) The acceptor of the second two electrons in the Q-cycle is cytochrome *b* containing low (cyt *b_L_*) and high (cyt *b_H_*) potential hemes. This chain supplies electrons to the Q_i_-site at the matrix side of the IMM, where they reduce one ubiquinone (CoQ_10_) to semiquinone and then to ubiquinol. Complexes I and III (Q_o_-site) are the sources of superoxide (O_2_^•–^).

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
