# Peer review of "CoQ10 and Mitochondrial Dysfunction in Alzheimer’s Disease"

_antioxidants, 2024, doi:10.3390/antiox13020191_

Round 1

Reviewer 1 Report

Comments and Suggestions for Authors

The authors present a review on the possible role of coenzyme Q10 in the prevention of Alzheimer's, acting on oxidative stress and mitochondrial dysfunction.

It is an interesting topic and although there are some reviews that address the issue of mitochondrial dysfunction and oxidative stress in Alzheimer's disease, there are few reviews that involve coenzyme Q10 exclusively in this pathology, although it does in other pathologies such as Parkinson's disease or neurodegenerative diseases in general.

There are aspects that should be considered to improve the reading and interest of this publication.

Abstract: I believe that the objective of this review should be clearly stated. The abstract talks about the importance of mitochondrial dysfunction and oxidative stress in Alzheimer's disease, the importance of CoQ in the correct mitochondrial function and stress modulation, but it does not indicate what the objective of this review is.

The first paragraphs of section 2 (page 3, line 49-69), are a bit chaotic or difficult to follow, it talks about the importance of beta amyloid and tau protein in the development of the disease and then about multiple other factors involved in the development of this disease, including free radicals and apoptotic processes, but without establishing relationships between them, until the final paragraph. I think I should write it, following the criteria shown in the final paragraph of that introduction.

​ Page 3, line 73-74: “age-related neurodegeneration is influenced by environmental factors (age, diet, exercise, lifestyle, and cognitive reserve).” Age is not an environmental factor. Explain what you mean by cognitive reserve, I don't think it is an environmental factor.

Page 3, line 108-112: “Alzheimer's disease biomarkers are mainly sought and studied (i) neuroimaging (focused on structural and functional changes, decreased connectivity, hypometabolism, and pathological aggregates of Aβ and tau [36-39], (ii) proteomic and metabolomic [40], and (iii) oxidative stress [41], mitochondrial [42,43], and neuroinflammatory [44]”. This sentence is not clear, it is understood what is meant, but it is not well understood. What does what appears in parentheses (i), (ii), (iii) mean?

In general point 2 is difficult to follow, it should focus more clearly on mitochondrial dysfunction and oxidative stress in Alzheimer's and specifically on tau protein and amyloid beta. It confusingly mixes aging and Alzheimer's disease. It is true that age is an important factor in Alzheimer's disease, as in any neurodegenerative disease and in other diseases such as cancer, cardiovascular disease, etc. But, here the aging process is explained at the same time as the Alzheimer's process and that makes for confusing reading. As I have indicated, it should focus on Alzheimer's disease and explain what happens in this pathology. Also, he repeats many times the same causes that should be considered, he talks about mitochondrial dysfunctionality and then stress, but here he comes back again to dysfunctionality. It is very difficult to follow and should be rewritten.

The rest of the article is almost a review on the synthesis and functions of Coenzyme Q10, leaving only a small section, 4.2, to the study of Coenzyme Q10 and Alzheimer's disease.

I consider that the article gives little information on the effect of Coenzyme Q10 and Alzheimer's disease, it should focus more on the existing studies and the reason for the contradictions observed, as shown in the discussion and conclusion, and less on the pathology of Alzheimer's disease and on the functions and synthesis of Coenzyme Q10. Perhaps there is still too little data in this regard for a review.

I believe that the article would be improved by reducing the generality sections and increasing the points made in the discussion and related to the studies on CoQ10 and Alzheimer's disease and cognitive degeneration.

Author Response

We are submitting a revised review article, "CoQ10 and mitochondrial dysfunction in Alzheimer‘s disease" (Manuscript ID: antioxidants-2829075) for publication in Antioxidants. We would like to thank reviewers for their valuable comments and evaluation of the manuscript.

A list of changes to each point raised by reviewers

Journal name: Antioxidants

Manuscript ID: antioxidants-2829075

Type of manuscript: Review

Title: CoQ10 and mitochondrial dysfunction in Alzheimer‘s disease

Authors: ZdenÄ›k Fišar, Jana Hroudová

Reviewer 1:

Abstract: I believe that the objective of this review should be clearly stated. The abstract talks about the importance of mitochondrial dysfunction and oxidative stress in Alzheimer's disease, the importance of CoQ in the correct mitochondrial function and stress modulation, but it does not indicate what the objective of this review is.

Response:

  • The sentence was added to the abstract: "This review summarizes knowledge about the pathophysiology of AD, in which CoQ10 may play a significant role, with the aim of evaluating the perspective of pharmacotherapy of AD with CoQ10 and its analogues". The more detailed objective of this review was added to last paragraph in the Introduction.

The first paragraphs of section 2 (page 3, line 49-69), are a bit chaotic or difficult to follow, it talks about the importance of beta amyloid and tau protein in the development of the disease and then about multiple other factors involved in the development of this disease, including free radicals and apoptotic processes, but without establishing relationships between them, until the final paragraph. I think I should write it, following the criteria shown in the final paragraph of that introduction.

              Response:

  • The first paragraph of section 2 was rewritten.

Page 3, line 73-74: “age-related neurodegeneration is influenced by environmental factors (age, diet, exercise, lifestyle, and cognitive reserve).” Age is not an environmental factor. Explain what you mean by cognitive reserve, I don't think it is an environmental factor.

              Response:

  • "Age" was deleted in environmental factors. "Cognitive reserve" is a result of the lifestyle; therefore "cognitive reserve" was deleted from environmental factors.

Page 3, line 108-112: “Alzheimer's disease biomarkers are mainly sought and studied (i) neuroimaging (focused on structural and functional changes, decreased connectivity, hypometabolism, and pathological aggregates of Aβ and tau [36-39], (ii) proteomic and metabolomic [40], and (iii) oxidative stress [41], mitochondrial [42,43], and neuroinflammatory [44]”. This sentence is not clear, it is understood what is meant, but it is not well understood. What does what appear in parentheses (i), (ii), (iii) mean?

              Response:

  • The sentence was reworded and (i) (ii) (iii) were removed.

In general point 2 is difficult to follow, it should focus more clearly on mitochondrial dysfunction and oxidative stress in Alzheimer's and specifically on tau protein and amyloid beta. It confusingly mixes aging and Alzheimer's disease. It is true that age is an important factor in Alzheimer's disease, as in any neurodegenerative disease and in other diseases such as cancer, cardiovascular disease, etc. But, here the aging process is explained at the same time as the Alzheimer's process and that makes for confusing reading. As I have indicated, it should focus on Alzheimer's disease and explain what happens in this pathology. Also, he repeats many times the same causes that should be considered, he talks about mitochondrial dysfunctionality and then stress, but here he comes back again to dysfunctionality. It is very difficult to follow and should be rewritten.

              Response:

  • The entire chapter "2. Alzheimer's disease" has been edited. Information about the processes associated with aging has been omitted, and those related directly to Alzheimer's disease have been retained. Repetitive information has been removed. Figure 2 has been added, showing the link between amyloid beta and tau pathology and mitochondrial dysfunction.

The rest of the article is almost a review on the synthesis and functions of Coenzyme Q10, leaving only a small section, 4.2, to the study of Coenzyme Q10 and Alzheimer's disease.

I consider that the article gives little information on the effect of Coenzyme Q10 and Alzheimer's disease, it should focus more on the existing studies and the reason for the contradictions observed, as shown in the discussion and conclusion, and less on the pathology of Alzheimer's disease and on the functions and synthesis of Coenzyme Q10. Perhaps there is still too little data in this regard for a review.

              Response:

  • A systematic Review about CoQ10 and AD dementia was published recently (Jiménez-Jiménez et al., 2023), therefore the subsection “3.2. CoQ10 in AD pathophysiology" was written only briefly and the sentence "A systematic review and meta-analysis of studies measuring CoQ10 levels in AD patients and other dementias and a review of the therapeutic effects of CoQ10 in humans with AD was recently published" was added.
  • To subsection "3.2. CoQ10 in AD pathophysiology" a paragraph on dietary intake and CoQ10 bioavailability was added. Information was added that "therapeutic importance of CoQ10 in treatment of AD is demonstrated mainly by studies on animal models of AD", and that "studies that provide clinical and mechanistic data regarding the efficacy of CoQ10 in the treatment of AD-related cognitive decline are lacking.” We have added links to papers dealing with the bioavailability of CoQ10 and its effect on neuroprotection and cognition.

I believe that the article would be improved by reducing the generality sections and increasing the points made in the discussion and related to the studies on CoQ10 and Alzheimer's disease and cognitive degeneration.

              Response:

  • Chapter 2 has been shortened and information on studies looking at CoQ10 in Alzheimer's has been added (however, human studies are few).

Reviewer 2 Report

Comments and Suggestions for Authors

Overview of the manuscript
The manuscript is a review focused on the study of the involvement of  Coenzyme Q10 (CoQ10) in  Alzheimer's disease (AD). Following the hypothesis of oxidative stress and mitochondrial disfunction as important events in the AD onset, particularly the age-related AD onset, the authors develop their review by analysing and reporting studies that have analysing the relationships between CoQ10 and AD.

GENERAL COMMENT

The complexity of the research carried out on the relationships between AD and oxidative stress has produced a very high quantity of scientific paper, in particular on the enigmatic response of CoQ10 and AD. This fact makes a work focused on this topic interesting and important.

The review is well performed, and the topic adequately analysed. The bibliographic section is rich and adequately updated and offers readers an important point of reference.

SPECIFIC COMMENTS

Pag. 2, line 63: add references to the end of the paragraph.

Pag. 4, line 183: the relationships between mitochondrial disease and the development of AD pathology is an important issue of your work, I suggest introducing a figure or scheme to illustrate the related events

Pag. 5, line 197: “RONS”? correct or explain

Pag. 7, line 326-329: I suggest extending the observation on dietary intake and CoQ10 availability and adding references.

Pag. 10, line 476: correct the capital letter.

References

Pag. 18, ref 78: The reference is too old, change or delete it.

Pag. 18, ref 98: The reference is too old. I suggest deleting it and replacing it in the text with ref 99

Author Response

We are submitting a revised review article, "CoQ10 and mitochondrial dysfunction in Alzheimer‘s disease" (Manuscript ID: antioxidants-2829075) for publication in Antioxidants. We would like to thank reviewers for their valuable comments and evaluation of the manuscript.

A list of changes to each point raised by reviewers

Journal name: Antioxidants

Manuscript ID: antioxidants-2829075

Type of manuscript: Review

Title: CoQ10 and mitochondrial dysfunction in Alzheimer‘s disease

Authors: ZdenÄ›k Fišar, Jana Hroudová

Reviewer 2:

Pag. 2, line 63: add references to the end of the paragraph.

              Response:

  • References have been added.

Pag. 4, line 183: the relationships between mitochondrial disease and the development of AD pathology is an important issue of your work, I suggest introducing a figure or scheme to illustrate the related events

              Response:

  • Figure 2 has been added with a diagram of the main steps in the pathophysiology of Alzheimer's disease, focusing on the role of mitochondrial dysfunction in the development of the disease.

Pag. 5, line 197: “RONS”? correct or explain

              Response:

  • "RONS" was corrected to "ROS".

Pag. 7, line 326-329: I suggest extending the observation on dietary intake and CoQ10 availability and adding references.

              Response:

  • A paragraph was added about dietary intake and CoQ10 bioavailability in the brain (with 14 citations).

Pag. 10, line 476: correct the capital letter.

              Response:

  •  

References

Pag. 18, ref 78: The reference is too old, change or delete it.

Pag. 18, ref 98: The reference is too old. I suggest deleting it and replacing it in the text with ref 99

              Response:

  • Both old references have been deleted.

Round 2

Reviewer 1 Report

Comments and Suggestions for Authors

I consider that the questions raised have been answered correctly and the article has been improved.

If it is true that there are small points that could be modified, especially related to the information incorporated into the new article, in some points it should be better fitted in relation to the information that was already in the article.

For example: page 1 lines 29-33: “Pathophysiological biomarkers of brain aging include factors that are linked to the mitochondrial dysfunction [1]. Amyloid beta (Aβ) and tau-induced neurotoxicity, which is linked to mitochondrial dysfunction and oxidative stress, have been postulated to play critical roles in the pathophysiology of Alzheimer's disease (AD).” They seem like two phrases that are not related to each other, but they are.

Page 2 lines 53-58: “This review summarizes findings on the role of oxidative stress and mitochondrial dysfunction in the pathophysiology of AD, with a focus on the role of CoQ10. The aim of this review is to show that the understanding of the connection of specific Aβ and tau pathology in AD with mitochondrial dysfunction and oxidative stress, together with considering the function of CoQ10, give a potential for further research into the role of CoQ10 and its analogues in the treatment of AD.” It's repetitive.

These are two examples, the fit of the inserted text with the existing one must be reviewed.

Author Response

We are submitting a revised review article, "CoQ10 and mitochondrial dysfunction in Alzheimer‘s disease" (Manuscript ID: antioxidants-2829075) for publication in Antioxidants. We would like to thank reviewers for their valuable comments and evaluation of the manuscript.

A list of changes to each point raised by reviewers

Journal name: Antioxidants

Manuscript ID: antioxidants-2829075

Type of manuscript: Review

Title: CoQ10 and mitochondrial dysfunction in Alzheimer‘s disease

Authors: ZdenÄ›k Fišar, Jana Hroudová

Reviewer 1:

If it is true that there are small points that could be modified, especially related to the information incorporated into the new article, in some points it should be better fitted in relation to the information that was already in the article.

For example: page 1 lines 29-33: “Pathophysiological biomarkers of brain aging include factors that are linked to the mitochondrial dysfunction [1]. Amyloid beta (Aβ) and tau-induced neurotoxicity, which is linked to mitochondrial dysfunction and oxidative stress, have been postulated to play critical roles in the pathophysiology of Alzheimer's disease (AD).” They seem like two phrases that are not related to each other, but they are.

Page 2 lines 53-58: “This review summarizes findings on the role of oxidative stress and mitochondrial dysfunction in the pathophysiology of AD, with a focus on the role of CoQ10. The aim of this review is to show that the understanding of the connection of specific Aβ and tau pathology in AD with mitochondrial dysfunction and oxidative stress, together with considering the function of CoQ10, give a potential for further research into the role of CoQ10 and its analogues in the treatment of AD.” It's repetitive.

These are two examples, the fit of the inserted text with the existing one must be reviewed.

Response:

  • Page 1 lines 29-33: The first paragraph in the Introduction has been edited.
  • Page 2 lines 53-58: The last paragraph in the Introduction summarizes the objective of the Review. At the earlier request of reviewer1, the objective of the Review was added to the Abstract. However, due to the limited scope of the Abstract, the objective of the Review was presented only very briefly. Therefore, the more detailed objective of this review was added to the last paragraph in the Introduction. As we consider it important to state at the end of the Introduction objective of the Review, we have not removed the repetition in this paragraph.
  • Modifications were made throughout the text (additions, deletions, reformulations, moves) so that the presented information is connected to each other and does not repeat itself. In particular, the following subsections were modified: 2.1. Risk factors, 2.3. Mitochondrial hypothesis, 2.4. Oxidative stress hypothesis, 3. Coenzyme Q10, 3.2. CoQ10 in AD pathophysiology, and 4.2. CoQ10 in AD treatment. We believe that we have achieved better clarity and readability of the text.
